# PARylation During Transcription: Insights into the Fine-Tuning Mechanism and Regulation

**DOI:** 10.3390/cancers12010183

**Published:** 2020-01-11

**Authors:** Zoltán G. Páhi, Barbara N. Borsos, Vasiliki Pantazi, Zsuzsanna Ujfaludi, Tibor Pankotai

**Affiliations:** Department of Oral Biology and Experimental Dental Research, Faculty of Dentistry, University of Szeged, Tisza Lajos krt 83, H-6722 Szeged, Hungary; pahizol@gmail.com (Z.G.P.); vasopantazi@outlook.com (V.P.); odianya@bio.u-szeged.hu (Z.U.)

**Keywords:** transcription, PARylation, PARP, DNA damage, transcription silencing

## Abstract

Transcription is a multistep, tightly regulated process. During transcription initiation, promoter recognition and pre-initiation complex (PIC) formation take place, in which dynamic recruitment or exchange of transcription activators occur. The precise coordination of the recruitment and removal of transcription factors, as well as chromatin structural changes, are mediated by post-translational modifications (PTMs). Poly(ADP-ribose) polymerases (PARPs) are key players in this process, since they can modulate DNA-binding activities of specific transcription factors through poly-ADP-ribosylation (PARylation). PARylation can regulate the transcription at three different levels: (1) by directly affecting the recruitment of specific transcription factors, (2) by triggering chromatin structural changes during initiation and as a response to cellular stresses, or (3) by post-transcriptionally modulating the stability and degradation of specific mRNAs. In this review, we principally focus on these steps and summarise the recent findings, demonstrating the mechanisms through which PARylation plays a potential regulatory role during transcription and DNA repair.

## 1. Introduction

### The Mechanism of Poly-ADP-Ribosylation and the PARP Superfamily

PARylation is a reversible post-translational modification (PTM), in which writers, such as poly(ADP-ribose) polymerases (PARPs) as well as erasers, including poly(ADP-ribose) glycohydrolases (PARGs) and ADP-ribosyl hydrolase 3 (ARH3) are involved [1,2,3,4,5,6,7]. PARPs are NAD^+^-dependent enzymes and thus require a source of NAD^+^ which is provided by nicotinamide mononucleotide adenylyl transferases (NMNATs) [8]. ADP-ribosylation is a multistep process, involving initiation, elongation, branching, and the release of PAR units. First, PARP binds to nicotinamide adenine dinucleotide (NAD^+^) and cleaves the nicotine amid unit, catalysing the transfer of the ADP-ribose moieties to the acceptor protein [9]. During initiation, the first ADP-ribose monomer can be covalently linked to Lys, Arg, Glu, Asp, Cys, Ser, or Thr amino acid residues of the acceptor protein [9]. During the branch formation step, 2’-1’ ribose-ribose bonds are generated between ADP-ribose units.

In human cells, PARPs are classified based on their enzymatic activity: PARP1, PARP2, PARP5a, and PARP5b catalyse PAR chain formation, while PARPs 3, 4, 6–8, 10–12, and 14–16 have been described as mono-ADP-ribosyl transferases (MARTs) [9]. PARP5a and PARP5b share a high level of similarity and are also called tankyrase 1 and tankyrase 2, respectively, due to their ankyrin repeat region and the sterile alpha motif [10]. Since PARylation is a reversible process, the covalently-attached PAR can be removed by PARGs and ARH3 as well, keeping the PAR levels in the cell under control. While PARG can efficiently cleave the PAR O-glycosidic bond, ARH3 is mainly responsible for the hydrolysis of protein-free PAR [11].

## 2. PARylation in Transcription Regulation

### 2.1. The Major Regulatory Steps of Transcription Activation

RNA synthesis requires a well-coordinated regulation of transcription at different levels. As a first step of initiation, transcription factor II (TFII)D-TFIIA-TFIIB binds to the promoter region; then TFIIF, along with RNA polymerase II (RNAPII), is recruited, resulting in the stabilisation of the pre-initiation complex (PIC) [12]. Next, TFIIE and TFIIH join to the core PIC, contributing to its association with the mediator complex [13]. Xeroderma pigmentosum type B (XPB), one of the subunits of TFIIH, induces DNA unwinding around the transcription start site (TSS) and initiates the formation of the transcription bubble [14]. Subsequently, the cyclin-dependent kinase 7 (CDK7) subunit of TFIIH phosphorylates the C-terminal domain (CTD) of RNAPII at Ser5, which is indispensable for transcription initiation. Following the synthesis of approximately 20–60 base pairs of RNA, RNAPII is stopped, which is the so-called promoter-proximal pausing [15]. During this step, negative elongation factors, including the dephosphorylated form of DRB sensitivity inducing factor (DSIF) and negative elongation factor (NELF), bind to the RNAPII, thereby hindering the elongation [16,17]. The cyclin-dependent kinase 9 (CDK9) subunit of positive transcriptional elongation factor b (P-TEFb) promotes the elongation process by phosphorylating DSIF, NELF, and RNAPII CTD at Ser2 [18,19]. Consequently, the phosphorylated NELF complex dissociates, while DSIF becomes a positive elongation factor responsible for recruiting other factors, such as cyclin-dependent kinase 12 (CDK12), which can also catalyse the phosphorylation of RNAPII CTD at Ser2 and by this, supporting the elongation step [20,21].

### 2.2. PARP1 Plays a Key Role in the Fine-Tune Regulation of Transcription Initiation

During transcription initiation, PARP1 can PARylate sequence-specific transcription factors, such as nuclear factor kappa-light-chain-enhancer of activated B cells (NF-κB), Myb-related protein B (B-MYB), organic cation transporter 1 (OCT1), sex determining region Y-box 2 (SOX2), and Krueppel-like factor 8 (KLF8), as well as oestrogen- and retinoic acid receptors, which can either inhibit or enhance the activity of these factors (Figure 1A) [22,23,24,25,26,27,28,29]. Along with PARP1, PARP7 has been identified as an important player in the transcription regulation of pluripotency genes and in their protection from epigenetic repression [30]. Originally, PARP1 was identified as TFIIC, which is capable of facilitating the initiation steps of mRNA synthesis through direct interaction with the basal transcription machinery [31]. Recently, it has been shown that PARP1 can also serve as a scaffold protein by stimulating the recruitment of coregulator complexes, such as p300, NF-κB, and p50 as well as the mediator complex to promoter regions (Figure 1A) [32]. It has been proven that the DNA-binding, rather than the catalytic activity of PARP1, enhances transcription by promoting the early steps of PIC formation [33]. However, only a limited amount of data is available, which suggests that any member of the basal transcription machinery is PARylated [34]. Moreover, PARP1 facilitates not only the recruitment but also the release of specific transcription co-regulators, leading to dynamic exchange between transcription factors, such as TLE family member 1 (TLE1) transcriptional corepressor complex to histone-acetyltransferase (HAT)-containing complex in neurons (Figure 1A) [35]. Additionally, PARP1 is also necessary for maintaining the relaxed chromatin structure of actively transcribed genes, supporting the active transcription of these genes [36]. In contrast, biochemical studies have revealed that PARP1 is not indispensable for the initiation of transcription on intact DNA templates, but single-stranded DNA breaks (SSBs) promote PARP1 binding, resulting in repression of nick-dependent transcription [31].

### 2.3. PARP1 Mediates Promoter-Proximal Pausing and Transcription Elongation

Following transcription initiation, RNAPII promoter-proximal pausing can be observed between the +20 and +150 region around the TSS [37]. In the early elongation step, RNAPII activity is temporarily paused by the contribution of NELF-E and DSIF [21]. During the promoter-proximal pause release, P-TEFb phosphorylates DSIF, NELF-E, and the CTD of RPB1 (the largest subunit of RNAPII) at Ser2, making it competent for transcription elongation and preventing its inhibition by DSIF and NELF-E (Figure 1B) [37,38]. It has been also demonstrated that PARP1 directs RNAPII for PARylation shortly after DNA damage, facilitating the recruitment of NELF-E to RNAPII and by this, NELF-E plays a potential role in transcription silencing at the DNA break sites (Figure 1B) [39].

The *Drosophila* orthologue of PARP1 plays an important regulatory role during the elongation phase of heat shock-induced transcription [23,40]. Two members of the NELF complex, NELF-A and NELF-E, have been shown to be targeted by PARP1 [41]. Moreover, the phosphorylation of NELF-E catalysed by CDK9/P-TEFb, is indispensable for its subsequent PARylation, resulting in the attenuation of its DNA-binding ability (Figure 1B) [41]. Additionally, inhibition of either PARP1 or CDK9/P-TEFb results in reduced phosphorylation of RNAPII CTD at Ser2, thus leading to promoter-proximal pausing. The genome-wide distribution of PARylation is mostly enriched at actively transcribed regions, where high levels of NELF-B, RNAPII and H3K4me^3^ can be observed [41]. Based on these data, PARylation plays a potential regulatory role throughout the entire transcription process. Nevertheless, further investigations are needed to highlight the proper PARP-mediated regulatory mechanisms.

### 2.4. PARylation Regulates Transcription Responses during DNA Damage

Genome integrity is being constantly challenged by various genotoxic stresses, which can result in different types of DNA lesions, including SSBs and double-stranded breaks (DSBs). DNA damage repair requires tight regulation, since inappropriate repair can lead to genome instability and tumourigenesis. In this regard, eukaryotes have evolved various mechanisms addressing DSBs, including homologous recombination (HR) and non-homologous end-joining (NHEJ) [42]. Although different factors are involved in these DNA repair pathways, crosstalk may occur between them. When DSBs arise within an actively transcribed unit, NELF-E and -A are rapidly accumulated at the break sites through a PARP1-dependent manner, leading to the silencing of this transcription unit [39]. Additionally, in vitro PAR-binding assays have also revealed that NELF-E has elevated binding capacity to PAR moieties, and the interaction between PARP1 and NELF-E is weakened following exposure to ionising radiation [41]. Additionally, the broken DNA region is PARylated by PARP1, facilitating the recruitment of NELF-E and silencing of transcription at the break site [39]. Furthermore, at sites of laser-induced DNA damage, PARP1 can also indirectly regulate transcription through its interaction with the TIMELESS protein. It has been also established that the PARP1–TIMELESS complex plays an essential role in HR [43].

The dual inhibition of PARP1 and PARP2 can lead to reduced binding of nucleosome remodelling deacetylase (NuRD) complex to sites of DNA damage [44]. NuRD can facilitate the recruitment of protein kinase C-binding protein 1 (ZMYND8) to the PARylated DNA damage sites [45]. Lysine-specific demethylase 5A (KDM5A) plays a key role in demethylation of H3K4 to regulate the binding of ZMYND8-NuRD complexes to the DSB (Figure 2A,B) [46]. Consequently, PARP1 can regulate transcription silencing not only by recruiting chromatin remodellers but also demethylases, which can remove the methyl groups from H3K4me^3^, responsible for transcription activation. During DNA damage-induced transcription silencing, PARP1 facilitates the recruitment of the polycomb repressive complex 1 and 2 (PRC1 and PRC2), ensuring the proper chromatin structure and transcription arrest at the damaged sites [44,47]. Moreover, the chromodomain Y like (CDYL) protein interacts with PRC2 in a PARP-dependent manner [48,49]. During DSB-induced transcription silencing, PARP1 promotes the recruitment of KRAB-associated protein-1 (KAP-1), heterochromatin protein 1 (HP1), and suppressor of variegation 3–9 homolog 1 (SUV39H1) (Figure 3A) [50]. SUV39H1 methylates H3K9 around the DSB, leading to the recruitment of additional KAP-1/HP1/SUV39H1 complexes thereby contributing to the spreading of H3K9me^3^ signal (Figure 3A,B) [50]. It results in the activation of histone acetyltransferase 5 (KAT5), which acetylates ataxia-telangiectasia mutated (ATM) (Figure 3C) [51,52]. ATM phosphorylates KAP-1, leading to the dissociation of the KAP-1/HP1/SUV39H1 complex from the chromatin, which allows the activation of the ATM-dependent HR repair pathway (Figure 3C,D) [50].

By contrast, during NHEJ, the Ku70/Ku80 heterodimers bind to the broken DNA ends and recruit the DNA-dependent protein kinase catalytic subunit (DNA-PKcs), a key player in transcription silencing processes. Additionally, PARP1 has been shown to interact with both DNA-PKcs and Ku70/80 [53]. Furthermore, even a single DSB can lead to DNA-PKcs-dependent transcription silencing, suggesting that PARP1 could also be involved in this pathway. At the end of this process, the HECT E3 ubiquitin ligase, WWP2 directs the stalled RNAPII complex to proteasomal degradation [54].

A recent systematic analysis has demonstrated that the TFIID complex member, TBP-associated factor 15 (TAF15), is bound to laser-induced DNA damage sites in a PAR-dependent manner [55]. Moreover, a proteome-wide mass-spectrometry analysis has revealed that the RNA-binding protein, RNA-binding motif protein X-linked (RBMX), is PARylated following exposure to genotoxic stress [56,57]. The DNA damage-induced appearance of PARylated transcription factors suggests an uncommon and transient transcription bursting, which generates the so-called DDR RNAs (DDRNAs) or DSB-induced RNAs (diRNAs) [58]. Furthermore, PARylation can indirectly facilitate transcription silencing through diRNA-mediated chromatin compaction [58,59].

The diRNAs presumably contribute to the recruitment of DDR factors and chromatin modifiers or participate in transcription silencing at DNA break sites [60]. The presence of DDRNAs at the damage sites is critical for the activation of DDR, since in the case of hindering the formation of double-stranded RNAs by silencing either Drosha or Dicer, tumour suppressor P53-binding protein 1 (53BP1) foci formation is highly reduced. Additionally, it has been also shown that breast cancer type 1 (BRCA1), implicating in HR and competing with 53BP1 and DNA repair protein RAD51 homolog 1 (RAD51) foci formation, is dramatically reduced upon silencing Drosha regardless of the cell cycle phase [61].

This finding might be explained by the fact that in the absence of sister chromatids, the HR pathway can only be activated in the presence of an RNA strand, which is ensured by DNA damage induced *de novo* transcription [62]. On the contrary, the recruitment of DDRNAs is not influenced by 53BP1 but rather it is highly dependent on the presence of RNAPII. This suggests a potential role of RNAPII in DDR activation by synthesising damage-induced long non-coding RNAs (dilncRNAs) at the site of DNA damage. It seems that the broken DNA ends can serve as promoters for the transcription of dilncRNAs, while simultaneously with this process, the transcription of coding regions is prevented. The transcription of dilncRNA is bidirectional, since it can be initiated towards both directions from the break site. However, DNA:RNA hybrids should be resolved by helicases or RNase H enzymes, since these can prevent further recruitment of the repair factors to the site of DNA damage [63]. Recently, it has been demonstrated that besides the interaction with RNAPII, BRCA1 also contributes to the resolution of DNA:RNA hybrids and preserves genome integrity through the recruitment of DNA/RNA helicase senataxin (SETX) to the terminal regions of genes [64,65]. Since HR is the most accurate DSB repair pathway, the appropriate and controlled recruitment of BRCA1/2 to the damage site is indispensable for the efficient repair. At this point, the inhibition of PARPs can counteract with the ongoing repair processes, resulting in genome instability [66]. Additionally, in BRCA1/2-deficient tumour cells, higher sensitivity to PARP inhibition can be observed [67,68]. These results as well as the preclinical trials highlight the substantial role of PARP inhibitors in cancer therapy [67,69,70,71]. Additional factors, including ATP-dependent RNA helicase A (DHX9), PARP1, scaffold attachment factor B2 (SAFB2), multiple myeloma SET domain (MMSET) and DNA-PKcs have been recently identified as proteins that interact with RNA/DNA hybrids [72]. Furthermore, DHX9 along with PARP1 plays a remarkable role in preventing R-loop accumulation and facilitating transcription termination [72]. PARP1 contributes to the enhancement of DHX9 helicase activity [73]. DHX9 interacts with a large number of proteins related to transcription, including RNAPII, suggesting that DHX9 travels with the elongating RNAPII and contributes to the resolution of R-loops in a PARP1 dependent manner [72].

Moreover, it has been shown that topoisomerase I and II play an indispensable role in the RNAPII pause release [74]. Elongating RNAPII induces torsional and topological stresses in the super-helical DNA, which should be resolved by topoisomerase I and II [75,76,77,78]. Topoisomerases take part in the activation of DDR during transcription activation and elongation processes [74]. Topoisomerase I is involved in the prevention of R-loop formation, which is one of the major sources of transcription-coupled genome instability, by removing negative supercoiling structures behind the RNAPII [78,79]. Upon transcription blockage, spliceosome displacement could result in R-loop formation, leading to ATM activation in a DSB-independent manner [80,81]. As a consequence of RNA/DNA hybridisation, ssDNA strands are formed, which are more susceptible to different kinds of DNA damage. Additionally, the formation of R-loops can lead to genome instability by interfering with DNA replication [82,83]. In addition to topoisomerases, R-loop formation can also be inhibited by RNase H, RNA/DNA helicases, and suppressors of proteins promoting R-loop formation [84,85]. Although Topoisomerase I is responsible for alleviating the torsional stress in the DNA, it gets trapped and is accumulated in close proximity to the DNA lesions. This leads to failures in DNA repair, genome instability, and tumourigenesis [86]. Such phenomenon occurs during camptothecin (CPT)-induced topoisomerase I inhibition, resulting in the accumulation of antisense RNAPII transcripts and then R-loop formation at actively transcribed regions [87,88,89]. Following CPT treatment, PARP1 interacts with topoisomerase I in both nucleolar compartments, playing a role in eliminating covalent topoisomerase I–DNA complexes through PARylation and recruiting repair factors to these sites [90,91]. In addition to PARP1, through the PARylation of topoisomerase I, PARP2 also plays a pivotal role in the removal of the stalled enzyme. By this mechanism, both PARP1 and PARP2 have a positive impact on preserving genome stability. Furthermore, PARP enzymes interfere with the actions of CPT, resulting in drug resistance. Therefore, combining PARP inhibitors with CPT may enhance therapeutic efficacy [92].

## 3. PARylation in the Regulation of DNA Damage-Induced Chromatin Structural Changes

Several studies suggest that both reduced structural constraints and altered nucleosome occupancy influence the accessibility of chromatin in response to DNA damage. Additionally, as a consequence of persistent DNA damage, the unfolding and spatial expansion of certain chromatin regions can be observed. Following DNA damage, robust chromatin decondensation occurs in a PARP-dependent manner [93,94]. The addition of the highly negatively charged PAR chains to histones and to other chromatin-associated proteins results in an electrostatic repulsion with the negatively charged DNA, leading to chromatin relaxation. Following DNA damage, PARP1 can also initiate chromatin conformational changes through interaction with other chromatin-modifying factors. Therefore, the activity of PARP1 seems to be indispensable for chromatin decondensation, contributing to initiation of DDR signalling and the recruitment of repair factors [55,95]. Following laser-micro irradiation, PARP1 is recruited to the site of DNA damage within seconds, while PARP2 binds only 30 s later. This finding supports that PARP1 is mainly responsible for the transient reorganisation of the chromatin structure [96,97]. Furthermore, kinetic analyses have shown that the binding of PARP1 is necessary for the recruitment of MRE11–RAD50–NBS1 (MRN) to the DSB sites (Figure 4A) [96]. Following DSB recognition, ATM, recruited by the MRN complex, phosphorylates H2A.X at S139 (referred to as γH2A.X), which triggers the recruitment of ring finger protein 8 (RNF8) and ring finger protein 168 (RNF168) ubiquitin ligases, participating in the K63-linked polyubiquitylation of H1 histones and in K13 and K15 ubiquitylation of H2A histones, respectively [98,99]. PARP1 facilitates the recruitment of the SWItch/sucrose non-fermentable (SWI/SNF)-related matrix-associated actin-dependent regulator of chromatin subfamily A member 5 (SMARCA5/SNF2H) to the sites of DNA damage and promotes the interaction between SMARCA5 and ADP-ribosylated RNF168 (Figure 4B) [100]. These results have confirmed that PARylation is a crucial step both for chromatin reorganisation and RNF168-mediated ubiquitylation of H2A, being responsible for the recruitment of additional DDR factors [100]. Hence, a functional link can be recognised between PARylation and ubiquitylation during DNA repair.

Subsequently to the auto-activation of PARP1, amplified in liver cancer 1 (ALC1), which interacts with PARP1 and histones, is recruited via a similar kinetic to PAR and catalyses nucleosome sliding in an ATP-dependent manner [101,102,103]. This recruitment requires its C-terminal PAR-binding macrodomain, which recognises PARylated PARP1. Recently, it has been shown that the interaction between the ATPase catalytic domain and the C-terminal macrodomain of ALC1 is necessary to keep ALC1 in an inactive state under physiological conditions. However, the activation of PARP1 disrupts the interaction of the aforementioned domains and allows the stimulation of the remodelling and PAR-dependent binding activity of ALC1 [104,105]. Nevertheless, the interaction of ALC1 with histones is dependent on the presence of the histone chaperone aprataxin and PNK-like factor (APLF), which are localised at the site of DNA lesion in a PAR-dependent manner and are PARylated by PARP1 [106,107]. It has been still unclear how the ALC1 contributes to the relaxation of the chromatin structure around the break site; therefore, further investigations are needed to reveal novel chromatin-associated interaction partners of it.

Additionally, a strong interaction has been demonstrated between APLF and macroH2A.1.1 following hydrogen peroxide-induced DNA damage, thus indicating a potential role of APLF in chromatin rearrangement [107]. Furthermore, this histone chaperone recognises branched PAR chains, catalysed by PARP2, and mediates histone H3 removal during DNA repair [97]. With regards to histone H3, the incorporation of the histone variant H3.3 has been linked to the PARP1-mediated accumulation of the chromodomain helicase DNA binding protein 2 (CHD2), triggering chromatin expansion (Figure 4B) [108]. Although CHD2 cannot recognise PAR moieties, earlier PAR-dependent events, including the rapid localisation of ALC1 and relaxation of the chromatin structure can trigger the accumulation of CHD2 near DNA breaks.

Several studies have highlighted that transient chromatin relaxation precedes chromatin compaction for protecting regions around the DNA break, resulting in the reveal of the DSB site that needs to be restored by the contribution of DDR factors [109]. Another hypothesis concerns the inhibition of replication and transcription at DSB-flanking regions in order to prevent interference with the repair machinery and mediate faithful repair. MacroH2A acts as a tumour suppressor and is a significant player in the maintenance of the heterochromatic structure as well as in the inactivation of the X chromosome, during which macroH2A inhibits the enzymatic activity of PARP1 [110,111,112,113]. The PAR-binding ability of the macrodomain modules has been recently demonstrated, underlining the PAR-capping of macroH2A.1.1, its capability of sensing PARP1 activation, and the subsequent reorganisation of chromatin structure by establishing a compacted chromatin environment [114]. Furthermore, only macroH2A.1.1 suppresses PARP1 activity, preventing the formation of an open chromatin structure. By contrast, all three histone variants of macroH2A (macroH2A.1.1, macroH2A.1.2 and macroH2A.2) retain the ability to stabilise condensed chromatin structure via their common linker region [115]. Subsequently, macroH2A.1.1 participates in recognizing and binding PAR chains to inactivate PARP1 (Figure 4C), whereas the linker region, being present in all three isoforms, may play an additional role in chromatin compaction. Together with PARP1 inhibition, the linker region contributes to the stabilisation of this architecture.

Factors involved in the NuRD complex, along with members of the Polycomb complex, play an indispensable role in the PARP1-associated DDR [44,55,116]. Particularly, metastasis associated protein 1 (MTA1), chromodomain helicase 3 and 4 (CHD3, CHD4), and all members of the NuRD complex are rapidly PARylated and exhibit enhanced binding to the site of the DSB. At laser-micro-irradiated sites, PARP1-mediated complete loss of transcription can be observed [44]. While Polo and colleagues revealed similar recruitment kinetics for CHD4 and PAR, other groups demonstrated that CHD3 and CHD4 require the initial PAR-dependent chromatin relaxation, potentially mediated by ALC1. Consequently, their accumulation appears at a later time point [116]. Recruitment of polycomb group ring finger 2 (PCGF2), polycomb group ring finger 4 (BMI1), and components of the polycomb repressive complex 1 (PRC1) are largely abrogated upon inhibition of PARP1. Interestingly, enhancer of zeste homolog 2 (EZH2), belonging to the polycomb repressive complex 2 (PRC2) and associated with transcription repression (H3K27me3 formation), is recruited depending on the enzymatic activity of PARP1 (Figure 4C) [44,47]. Recent findings have indicated that PARP1 inhibits the histone methyltransferase activity of EZH2, thus resulting in a more relaxed chromatin structure at the damaged regions [117,118]. Nevertheless, these opposing observations may arise from different kinds of DNA damage source, inducing different repair pathways or even from diverse phases of the repair process.

## 4. Role of PARP1 in RNA Metabolism

In addition to transcription, the abundance and decay of mRNAs can be post-transcriptionally regulated by controlling splicing, polyadenylation and nuclear export. A recent study has demonstrated that PARP1 is recruited to specific nucleosomes localised at the exon/intron boundaries, corresponding to specific splice sites. Additionally, PARP1 inhibition results in changes in the alternative splicing. Emerging data have indicated that PARP1 stimulates the recruitment of U2 snRNPs (small nuclear ribonucleoproteins), therefore positively influencing the exon recognition and the further splicing procedure [119]. Moreover, heterogeneous nuclear riboproteins (hnRNPs) can tightly bind to PAR chains, promoting the dissociation of hnRNPs from RNA and subsequent intron splicing [120]. So far, 11 human hnRNP proteins have been demonstrated to be capable of recognising and binding to PARylated targets. Affinity-purification mass spectrometry assays (AP-MS) combined with Gene Ontology classifications (GO) have shown that not only PARP1, but also PARP2 plays a role in intron splicing and interacts with hnRNPs [121]. It has been also shown that PARP10 is able to PARylate specific RNA substrates, leading to the protection of RNA ends which can act as a platform for recruiting other proteins [122]. Following stress responses, auto-PARylated PARP1 can bind to several nuclear proteins and initiate their transport to Cajal bodies, contributing to the regulation of either the assembly or disassembly of transcription- and splicing-related complexes [123]. Additionally, genome-wide data suggest that PARP1 can oppositely influence the transcription elongation by altering the elongation speed [124]. PARP1 also affects the assembly of human pre-mRNA 3′-processing complex. During transcription termination, PARylation is involved in hindering polyadenylation by catalysing the ADP-ribosylation of polyadenylate-polymerase (PAP), resulting in its reduced binding to mRNA transcripts [125]. Furthermore, PARP14 seems to be involved in the posttranscriptional regulation of mRNA stability since it can promote the degradation of specific transcripts by interacting selectively with tristetrapolin (TTP) [126]. Finally, PARylation could affect transcription by regulating the transport of specific mRNAs, since the nuclear export of mature mRNAs is also regulated by PARP1. In lipopolysaccharide-treated cells, PARP1-dependent PARylation of embryonic lethal abnormal vision-like 1 (ELAV-like protein 1) triggers the RNA nuclear-cytoplasmic shuttling, leading to the enhanced stability of mRNA [127]. These data suggest that following stress responses, PARP enzymes could affect the mRNA maturation at multiple levels. PARylation may be involved in most of the RNA metabolism-related processes, such as splicing, polyadenylation and mRNA maturation. Nonetheless, further investigations are needed to the more precise understanding of these regulatory pathways.

## 5. Discussion

In addition to DNA damage response, PARylation regulates various processes, such as chromatin remodelling, transcription activation and repression, ubiquitylation, RNA metabolism as well as cellular stress responses. PARylation can have the following effects on these processes: (1) it can ensure a surface for protein interactions, and (2) certain proteins, such as Imitation SWI (ISWI) and ALC1, possessing a PAR-binding domain (consisting of a PAR-binding zinc finger (PBZ), a PAR-binding motif (PBM), and a WWE), can be recruited to DNA through this process. Additionally, PARP1 can act as a scaffold protein by regulating the recruitment of transcription co-regulator complexes, such as p300 or the mediator complex. These results also support that PARylation is implicated in various mechanisms by promoting complex assembly [32]. On the other hand, during DNA repair, PARPs have an indispensable function in identifying DNA breaks and participating in DNA repair pathway choice. In the absence of PARylation, the insufficient activation of proteins involved in DNA repair can result in the malfunction of the repair mechanism. Furthermore, inappropriate activation of any DNA repair pathway can contribute to genome instability, leading to tumourigenesis. For instance, in HR-deficient tumour cells, in the absence of BRCA1/BRCA2, PARP1 inhibition has been shown to have cytotoxic side-effects. The putative mechanism of this hypersensitivity can be explained by the fact that PARP inhibitors can disturb the recruitment of BRCA1 to the damaged sites, resulting in inadequate HR activation. Although PARP1 is a well-characterised protein, the precise function of other PARPs in regulating other cellular processes has remained unclear. PARP inhibitors may alleviate the speed of DNA repair, leading to the collapse of the replication fork and high therapeutic efficacy during tumour therapy [71]. Moreover, PARylation can interfere with the early recruitment of both BRCA1 and BRCA2, contributing to HR deficiency [128,129,130,131]. Interestingly, during HR, PARPs not only recruit MRE11 and NBS1 to the damage sites, but also play a regulatory role during transcription. In HR-related de novo transcription, DHX9 interacts with PARP1, thereby regulating the transcription [72,73]. Moreover, BRCA1-RNAPII interaction contributes to the resolution of DNA:RNA hybrids [64,65]. These results highlight that PARPs can act as transcription regulators in various processes, which can reveal new possibilities in applying PARP inhibitors in clinical trials [132,133]. Although several clinical reports have already demonstrated that PARP inhibitors could be beneficial during tumour therapy, we have to mention that the exact biochemical mechanisms regulated by these PARP inhibitors still remained unexplored. Therefore, further investigations are required to uncover these PARylation-mediated mechanisms to reduce the off-target effects of PARP inhibitors. 

## 6. Conclusions

In this review, we address the role of PARylation to understand their function during the transcription-coupled cellular responses. We summarise the possible canonical mechanisms by which PARylation exerts its regulatory roles during the transcription responses. However, despite the increasing knowledge on the related topic, the extent of the contribution of PARylation needs to be elucidated. Depending on the cellular context of this PTM, it can exert opposite effects on the same cellular processes. During transcription, it was shown that PARylation can determine the transcription state either by activating or inhibiting the transcription of different sets of genes, leading to distinct biological outcomes. According to this, it is worthwhile performing studies able to address the effect of PARP inhibition on transcription responses. Further efforts should be initiated for the better understanding of the underlying mechanism of actions to achieve more effective therapeutic benefits with minimal side-effects.

## Figures and Tables

**Figure 1 cancers-12-00183-f001:**
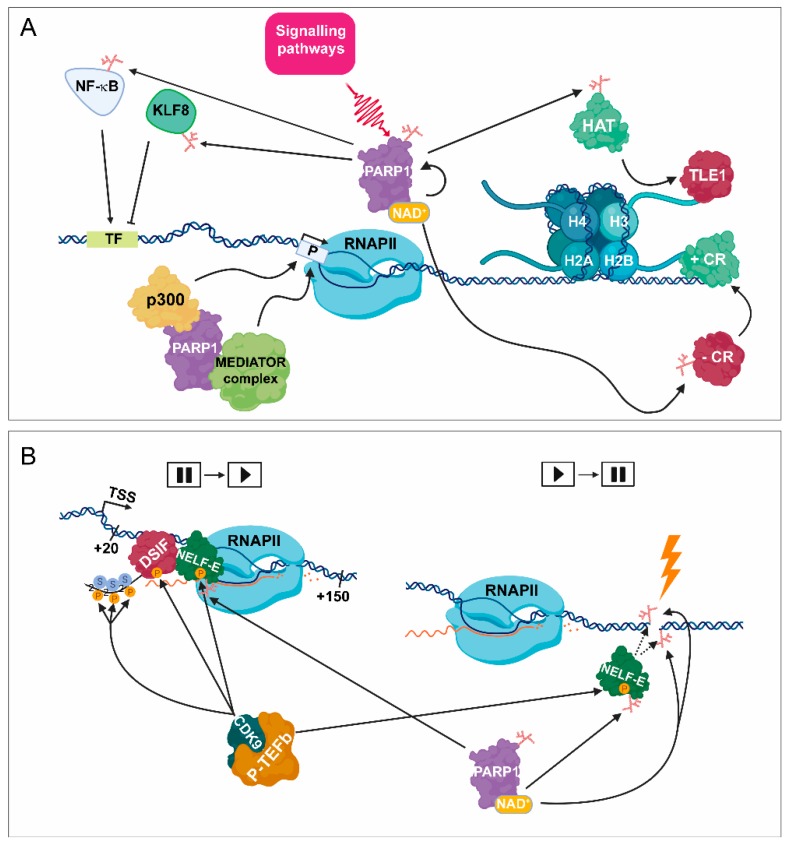
PARylation in transcription initiation. (**A**) As a consequence of the activation of certain signalling pathways, PARP1 catalyses the transfer of ADP-ribose through its binding to NAD^+^ cofactor. PARP1 can PARylate numerous sequence-specific transcription factors, such as NF-κB and KLF8, which can facilitate and attenuate transcription initiation, respectively. PARP1 can also act as a scaffold protein by promoting the recruitment of various co-regulator complexes, such as p300 and the mediator complex to the promoter region (P), leading to transcription initiation catalysed by RNAPII. Moreover, PARP1 is implicated in the release of subsequent corepressor complex, such as transducin-like enhancer protein 1 (TLE1), resulting in its exchange to the PARylated HAT complex. PARP1 also participates in the exchange between negative coregulators, (–CRs) to positive ones (+ CRs). TF = transcription factor binding site. (**B**) (Left part) Following transcription initiation, RNAPII is stopped at around +20–+150 bp from the TSS. The promoter-proximal pausing process is induced by negative elongation factors, such as NELF-E and DSIF. During the release of the initial pausing, CDK9/ PTEFb phosphorylates NELF-E, DSIF, and the CTD of RPB1 (the largest subunit of RNAPII) at Ser2. Phosphorylated NELF-E (P-NELF-E) is released from RNAPII, while phosphorylated DSIF (P-DSIF) acts as a positive elongation factor, and the S2P-RNAPII becomes capable of proceeding the elongation step. Right part: Following DNA damage, PARP1 targets P-NELF-E for PARylation, thereby hindering its DNA-binding ability, leading to transcription silencing.

**Figure 2 cancers-12-00183-f002:**
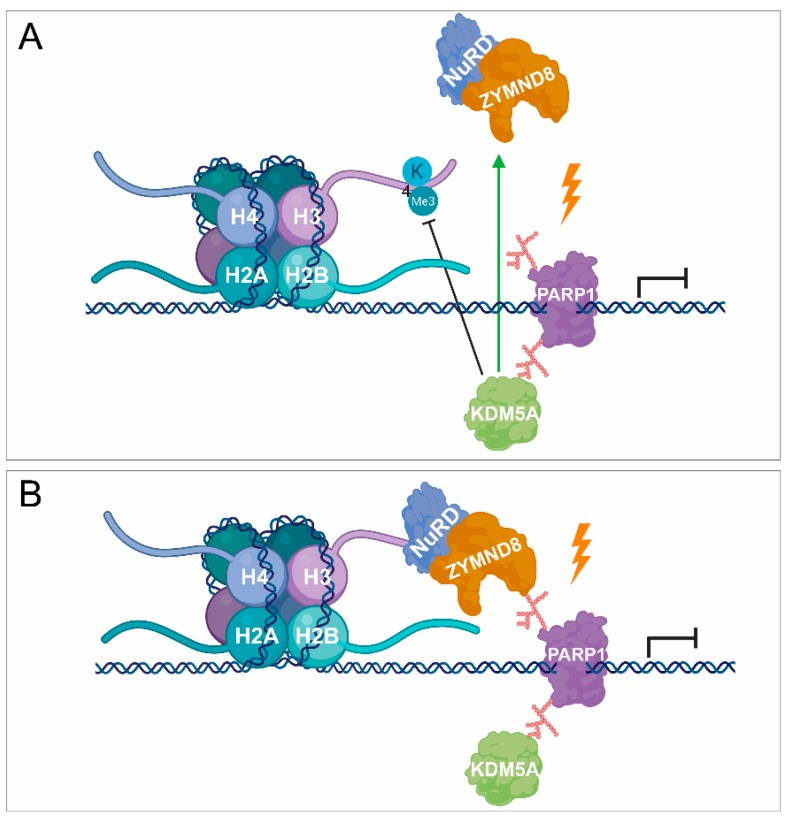
PARylation in transcription repression. (**A**,**B**) PARP1 is also involved in transcription silencing by recruiting demethylases, such as KDM5A, catalysing the removal of methyl groups and being responsible for transcription activation. KDM5A promotes the recruitment of NuRD and ZMYND8 to the lesion site by demethylating H3K4me^3^, thereby contributing to transcription silencing.

**Figure 3 cancers-12-00183-f003:**
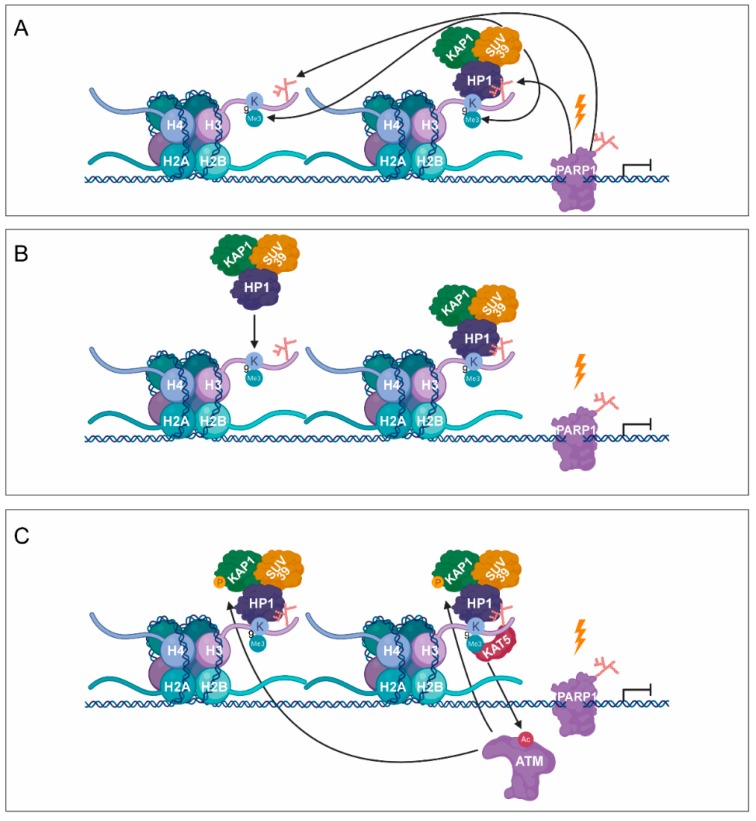
PARylation in transcription silencing during DNA damage. (**A**) As a response to DNA damage-induced transcription silencing, PARP1 facilitates the recruitment of SUV39H1, KAP1, and HP1. (**B**) Subsequent SUV39H1–HP1–KAP1 containing complexes are recruited, resulting in the spreading of H3K9me^3^ signal. (**C**) KAT5 is activated, which acetylates ATM, being responsible for the phosphorylation of KAP-1. (**D**) Phosphorylation of KAP1 contributes to the dissociation of SUV39H1–HP1–KAP1 from the chromatin.

**Figure 4 cancers-12-00183-f004:**
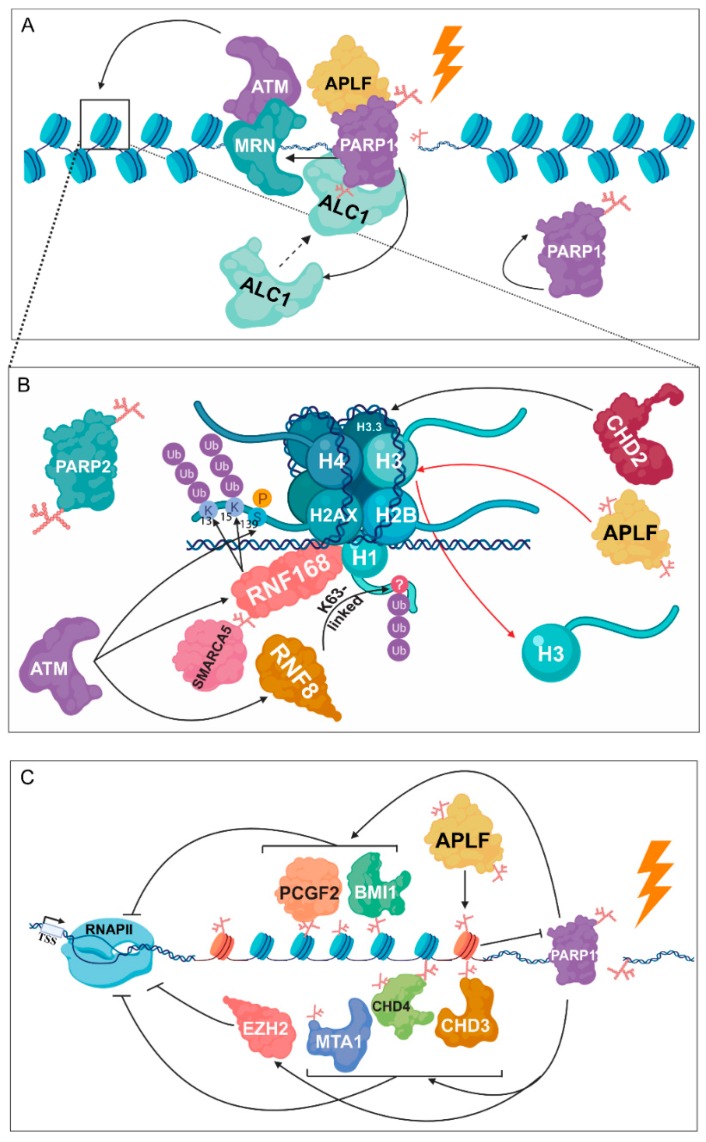
PARylation is required for chromatin structural changes following DNA damage. (**A**) PARP1 facilitates the binding of the MRN complex, which is known as the first DNA damage sensor, being responsible for the recruitment of ATM. Following PARP1 auto-activation, amplified in liver cancer 1 (ALC1) is implicated in nucleosome sliding by interacting with histones in a PARP- and aprataxin and PNK-like factor (APLF)-dependent manner. (**B**) ATM phosphorylates H2A.X at S139, resulting in the recruitment of RNF8 and RNF168. RNF8 catalyses the K63-linked poly-ubiquitylation of H1, while RNF168 is responsible for the ubiquitylation of H2A at K13 and K15. PARP1 promotes the recruitment of SMARCA5 to DSB sites and its subsequent interaction with RNF168. PARylated APLF, recognizing branched PAR chains catalysed by PARP2, participates in H3 removal during DNA repair. Furthermore, PARP1-mediated accumulation of chromodomain helicase DNA binding protein 2 (CHD2) leads to dynamic exchange of H3 to H3.3. (**C**) Members of the NuRD and polycomb complex, involving metastasis associated protein 1 (MTA1), CHD3, CHD4, polycomb group ring finger 2 (PCGF2), polycomb group ring finger 4 (BMI1), and enhancer of zeste homolog 2 (EZH2) take part in transcription inhibition through a PARP-mediated pathway.

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
