# Peer review of "PARylation During Transcription: Insights into the Fine-Tuning Mechanism and Regulation"

_cancers, 2020, doi:10.3390/cancers12010183_

Round 1

Reviewer 1 Report

This review manuscript covers the involvement of PARylation during transcription. As PARP inhibitors are being used in cancer treatment, this topic is important. A few suggestions for the authors to improve the manuscript:

1, many places reviews are used in stead of original research articles. For example, PARylation of transcription factors [16-18].

2, figures are generally too small and difficult to read.

3, section 2.3, "including DNA damage response (DDR), Homologous Recombination (HR) and Non-Homologous End-Joining (NHEJ)". This is inaccurate as DDR includes repair.

4, section 3, "kinetic analyses have shown that...., which binding is strongly dependent on PARP1". This is confusing, please rephrase.

5, the discussion section focused too much on the role of PARP in DDR, as the emphasis should be PARylation on transcription.

Author Response

We thank the Reviewers for reading and commenting on our manuscript. We have modified it according to their suggestions and we hope they will find our responses satisfying and the improved manuscript is acceptable for publication in the Cancers. Please find our point-by-point responses below:

Referee1:

1) Many places reviews are used instead of original research articles. For example, PARylation of transcription factors [16-18]

We have replaced most of the references to the original research article throughout the paper.

2) Figures are generally too small and difficult to read.

We thank the Reviewer for pointing out this important question. In the revised manuscript we increased the size of all figures and we also separated Figure 1 into 3 different figures (Figure 1-3) in order to improve their quality and resolution.

3) section 2.3, "including DNA damage response (DDR), Homologous Recombination (HR) and Non-Homologous End-Joining (NHEJ)". This is inaccurate as DDR includes repair.

Following the Reviewer’s recommendation, we modified the text accordingly:

‘In this regard, eukaryotes have evolved various mechanisms addressing DSBs, including Homologous Recombination (HR) and Non-Homologous End-Joining (NHEJ) [31].’

4) section 3, "kinetic analyses have shown that...., which binding is strongly dependent on PARP1". This is confusing, please rephrase.

We thank the Reviewer for pointing out this issue. In the revised version of the manuscript, we modified the text. ‘Furthermore, kinetic analyses have shown that the binding of PARP1 is necessary for the recruitment of MRE11-RAD50-NBS1 (MRN) to the DSB sites (Figure 4A) [78].’

5) the discussion section focused too much on the role of PARP in DDR, as the emphasis should be PARylation on transcription.

Based on the Reviewer’s recommendation, we modified the discussion session and we tried to emphasize the function of PARylation during transcription. Additionally, we included a new Conclusion session for our manuscript as requested.

Reviewer 2 Report

This review paper does a great job of summarizing what is currently known about PARylation in the context of transcriptional regulation. Figure 1, shows a thorough and straightforward schematic on the role of PARPs in catalyzing the PARylation of DNA repair and transcription factors. Figure 2 nicely demonstrates how PARP1 functions to remodel chromatin during DNA damage. Overall, the flow of the paper is clear and logical, it shows how PARylation is involved in transcription activation and how PARP1 is involved in modulating RNA metabolism.

Minor revisions:

The abstract mentions "post-translational modifications" but does not include the acronym, yet the acronym "PTM" is used in the introduction without a previous explanation on what PTM is.

In section 2.1, there is a small grammatical error, the authors state "It has been proved that..." -change "proved" to "proven"

In the discussion section, the clinical significance of PARP inhibitors is stated, yet no other mention of PARP inhibitors (in a clinical context) is made throughout the paper.

An earlier introduction to the clinical use of PARP inhibitors (in both cancer and inflammation) would be helpful, as it would:

-Reinforce the clinical relevance of understanding PARP/PAR biology

-Complement the point made (in the discussion section) regarding the "biochemical mechanisms" affected by PARP inhibition

Author Response

We thank the Reviewers for reading and commenting on our manuscript. We have modified it according to their suggestions and we hope they will find our responses satisfying and the improved manuscript is acceptable for publication in the Cancers. Please find our point-by-point responses below:

Referee2:

1) The abstract mentions "post-translational modifications" but does not include the acronym, yet the acronym "PTM" is used in the introduction without a previous explanation on what PTM is.

We thank the Reviewer for pointing out this issue. In the revised version of the manuscript, we added the acronym into the text.

2) In section 2.1, there is a small grammatical error, the authors state "It has been proved that..." -change "proved" to "proven"

We modified the text accordingly.

3) In the discussion section, the clinical significance of PARP inhibitors is stated, yet no other mention of PARP inhibitors (in a clinical context) is made throughout the paper. An earlier introduction to the clinical use of PARP inhibitors (in both cancer and inflammation) would be helpful, as it would. Reinforce the clinical relevance of understanding PARP/PAR biology. Complement the point made (in the discussion section) regarding the "biochemical mechanisms" affected by PARP inhibition.

By following the Reviewer’s recommendation, we included a chapter at page 7 in order to emphasize the importance of clinical significance and the mechanism of action of PARP inhibitors.

‘Recently, it has been demonstrated that besides the interaction with RNAPII, BRCA1 also contributes to the resolution of DNA:RNA hybrids and preserves genome integrity through the recruitment of DNA/RNA helicase Senataxin (SETX) to the terminal regions of genes [53,54]. Since HR is the most accurate DSB repair pathway, the appropriate and controlled recruitment of BRCA1/2 to the damage site is indispensable for the efficient repair. At this point, the inhibition of PARPs can counteract with the ongoing repair processes, resulting in genome instability. Additionally, in BRCA1/2-deficient tumour cells, higher sensitivity to PARP inhibition can be observed [55,56]. These results as well as the preclinical trials highlight the substantial role of PARP inhibitors in cancer therapy [55,57–59]’

Reviewer 3 Report

In this manuscript, Páhi et al. discuss the role of PARylation in transcriptional regulation. The review is substantially detailed and comprehensive, only minor points should be addressed in a revised form.

the flow of the text would be better if PARPs would be discussed first then a transition to transcription itself. Besides, the PARP chapter (currently 1.2) is a little bit too concise, and there is one sentence what is a bit confusing:

„Poly-ADP-ribosylation (PARylation) is a reversible PTM, catalysed by Poly(ADP-ribose) polymerases (PARPs) and Poly(ADP-ribose) Glycohydrolases (PARGs)”

The role of PARG suggested here is not correct as PARG does not catalyse the synthesis of the polymer, instead the opposite reaction, as it is mentioned in the last sentence of the chapter. In the hydrolysis of PAR, ADP-ribosyl hydrolase 3 (ARH3) also takes part so it has to be mentioned in the text.

The authors should point out in this chapter that PARP-1 is not the only PARP involved in transcriptonal regulation, but PARP-2, PARP-7, PARP-10 and PARP-14 have all been associated with transcription.

Chapter 1.1 should be included somewhere in the text in the second main chapter (2. PARylation in transcription regulation), not as a separate chapter

There are papers that have to be referenced in the text at due place, as they report very important findings in regard of the role of PARylation in transcription:

Caron et al., 2019 in Nat. Commun. https://doi.org/10.1038/s41467-019-10741-9. and

Wright et al., 2016 in Science doi: 10.1126/science.aad9335.  

Author Response

We thank the Reviewers for reading and commenting on our manuscript. We have modified it according to their suggestions and we hope they will find our responses satisfying and the improved manuscript is acceptable for publication in the Cancers. Please find our point-by-point responses below:

Referee3:

1) the flow of the text would be better if PARPs would be discussed first then a transition to transcription itself. Besides, the PARP chapter (currently 1.2) is a little bit too concise, and there is one sentence what is a bit confusing. Chapter 1.1 should be included somewhere in the text in the second main chapter (2. PARylation in transcription regulation), not as a separate chapter.

We agree with the Reviewer’s critique and we changed the order of the chapters. The revised version starts with the PARP introduction and the general transcription chapter now takes place at the beginning of chapter 2 (PARylation in transcription regulation).

2) The role of PARG suggested here is not correct as PARG does not catalyse the synthesis of the polymer, instead the opposite reaction, as it is mentioned in the last sentence of the chapter. In the hydrolysis of PAR, ADP-ribosyl hydrolase 3 (ARH3) also takes part so it has to be mentioned in the text. The authors should point out in this chapter that PARP-1 is not the only PARP involved in transcriptional regulation, but PARP-2, PARP-7, PARP-10 and PARP-14 have all been associated with transcription.

We thank the Reviewer for pointing out this issue. In the revised version of the manuscript, we modified the text. We also included some literature data according to the aforementioned PARPs (PARP2, PARP7, PARP10 and PARP14).

3) There are papers that have to be referenced in the text at due place, as they report very important findings in regard of the role of PARylation in transcription.

By following the Reviewer’s recommendation, the new references were implicated into the manuscript.

Round 2

Reviewer 1 Report

The authors addressed my previous concerns. The revised manuscript should provide the field with a good summary of how PARPs are involved in transcription, as well as the repair-transcription association.